# Portal Hypertension in Malnutrition and Sarcopenia in Decompensated Cirrhosis—Pathogenesis, Implications and Therapeutic Opportunities

**DOI:** 10.3390/nu16010035

**Published:** 2023-12-21

**Authors:** Ryma Terbah, Adam Testro, Paul Gow, Avik Majumdar, Marie Sinclair

**Affiliations:** 1Liver Transplant Unit, Austin Health, 145 Studley Road, Heidelberg, VIC 3084, Australia; ryma.terbah@austin.org.au (R.T.); adam.testro@austin.org.au (A.T.); paul.gow@austin.org.au (P.G.); avik.majumdar@austin.org.au (A.M.); 2Department of Medicine, The University of Melbourne, Parkville, VIC 3050, Australia

**Keywords:** cirrhosis, portal hypertension, sarcopenia, malnutrition

## Abstract

Malnutrition and sarcopenia are highly prevalent in patients with decompensated cirrhosis and are associated with poorer clinical outcomes. Their pathophysiology is complex and multifactorial, with protein-calorie malnutrition, systemic inflammation, reduced glycogen stores and hormonal imbalances all well reported. The direct contribution of portal hypertension to these driving factors is however not widely documented in the literature. This review details the specific mechanisms by which portal hypertension directly contributes to the development of malnutrition and sarcopenia in cirrhosis. We summarise the existing literature describing treatment strategies that specifically aim to reduce portal pressures and their impact on nutritional and muscle outcomes, which is particularly relevant to those with end-stage disease awaiting liver transplantation.

## 1. Introduction

The development of clinically significant portal hypertension in cirrhosis, defined by a hepatic venous pressure gradient (HVPG) of ≥10 mmHg, is associated with increased morbidity and mortality. The risk of clinical decompensation of liver disease, such as ascites, variceal haemorrhage or encephalopathy, increases progressively with increasing severity of portal hypertension [1]. Such events are associated with markedly reduced survival, with a median patient survival of over 12 years in patients with compensated cirrhosis, as compared to approximately 2 years once decompensation occurs [2].

Malnutrition and sarcopenia are increasingly recognised complications of cirrhosis. They are reported in up to 70% of patients with cirrhosis depending on the diagnostic method employed and patient population [3,4]. The prevalence of sarcopenia increases with increasing severity of liver disease [5,6] but confers mortality risk independent of the severity of liver disease [5,7,8].

The pathophysiology of sarcopenia in cirrhosis is complex, with multiple contributors including protein-calorie malnutrition, reduced glycogen stores, hormonal imbalances and myostatin upregulation, many of which are well described in the literature [9,10,11]. Portal hypertension is responsible for several of the mechanisms contributing to sarcopenia, but the direct role it plays is less widely described. With active therapies now available to treat portal hypertension, understanding its impact on sarcopenia is crucial to allow clinicians to consider appropriate therapy.

This review summarises the existing literature that specifically links portal hypertension to sarcopenia and malnutrition and discusses existing evidence for potential treatment strategies.

## 2. Prevalence of Sarcopenia in Cirrhosis

Although the consensus definition of sarcopenia requires the presence of reduced muscle strength, mass and functional limitation [12], various techniques have been employed in the medical literature to estimate sarcopenia in cirrhosis, and these differences in methodology likely influence the variable prevalence ranging from 30 to 70% [3]. The prevalence also varies depending on the patient population studied, with males being affected more frequently than females (41.9% vs. 28.7%) when measures of muscle mass are used, and patients with alcohol-related liver disease having a significantly higher prevalence than those with other aetiologies of liver disease (49.6% vs. 33.4%) [13]. Sarcopenia is also more common in patients awaiting liver transplant, with a reported prevalence of 50% in male patients and 33% in female patients in one multicentre study of 396 patients listed for liver transplant in North America [14] and a total of 41% [15] to 47.8% [16] of waitlisted patients in other studies.

Existing estimates of sarcopenia include bedside tests such as mid-arm muscle circumference, handgrip strength and bioelectrical impedance; measures of frailty such as the short physical performance battery, the 6 min walk test and the Liver Frailty Index; and radiological tests such as an ultrasound of quadriceps muscle diameter or muscle psoas index, CT skeletal muscle index (SMI) at the third lumbar vertebrae, CT psoas muscle thickness, dual-energy X-ray absorptiometry (DEXA) appendicular lean mass, and MRI fat-free muscle area. The strengths and weaknesses of each diagnostic method have been well described elsewhere [17,18,19].

The North American guidance statement [20] recommends CT SMI as the gold standard for quantifying muscle mass in cirrhosis, but it remains controversial whether functional or structural measures are more useful for patient care, as functional measures have been found to be a better predictor of outcome in females [21].

## 3. Correlation between Sarcopenia and Portal Hypertension

Sarcopenia is more prevalent with progressive liver disease. A meta-analysis by Tantai et al. found a prevalence of 46.7% in patients with Child–Pugh C cirrhosis, compared to 37.9% in patients with Child–Pugh B and 28.3% in Child–Pugh A cirrhosis [13]. Although sarcopenia may be more common with higher Child–Pugh scores, it does not necessarily correlate with the severity of portal hypertension. Paternostro et al. found that the presence of sarcopenia did not correlate with absolute HVPG values; however, clinical decompensation was significantly higher in the patients with sarcopenia compared to the non-sarcopenic patients (83.1% vs. 67.5%) [22]. Likewise, in a subanalysis using HVPG, Kang et al. did not find a significant difference in the proportion of patients with non-clinically significant portal hypertension (defined as HVPG 6–9 mmHg), clinically significant portal hypertension (HVPG 10–19 mmHg) and extremely severe portal hypertension (HVPG ≥ 20 mmHg) in a cohort of patients with sarcopenia compared to those without sarcopenia [23]. Rodrigues et al. [24] found that adipopenia was inversely correlated with HVPG; however, sarcopenia was not. These data suggest that portal hypertension is not the only cause of sarcopenia in this cohort, and as per previous reports [9,25,26], the pathogenesis of sarcopenia is complex, with multiple contributing factors.

## 4. Clinical Impact of Sarcopenia in Cirrhosis

Sarcopenia can negatively impact the prognosis of patients with cirrhosis and portal hypertension and has been associated with increased prevalence of decompensated liver disease [22] including ascites [22,27,28], hepatic encephalopathy [29,30] and infection risk [28,31]. It has also been independently associated with impaired quality of life [32].

Sarcopenia is also associated with an almost two-fold higher risk of death [13], and its presence is a major independent risk factor for mortality independent of the severity of portal hypertension [27]. The median survival of cirrhotic patients with sarcopenia has been reported to be significantly lower than non-sarcopenic patients (19 vs. 34 months) [33]. Several studies have noted that the highest impact of sarcopenia on mortality is seen in patients with Child–Pugh A and B cirrhosis, or with lower MELD scores, rather than in patients with Child–Pugh C cirrhosis [31,34]. The reasons for this are not fully clear, however may be partly due to patients with Child–Pugh C cirrhosis being already at high risk of developing acute-on-chronic liver failure (ACLF) and thus the absolute additive risk of sarcopenia is less pronounced [35].

The presence of sarcopenia also has implications for patients waitlisted for liver transplantation, with sarcopenia being associated with increased pre- [15] and post-transplant mortality [36], prolonged intensive care unit [37] and overall hospital stays [38], increased post-transplant infections and sepsis [39], and increased healthcare costs [40].

## 5. Pathophysiology of Portal Hypertension

Blood pressure is determined by blood flow and resistance, as expressed by Ohm’s law. Portal hypertension therefore develops when there is increased blood flow and/or resistance through the portal vascular bed. In cirrhosis, there are both fixed and dynamic contributors to portal hypertension.

### 5.1. Causes of Increased Intrahepatic Vascular Resistance

Cirrhosis results in irreversible fixed structural changes and alterations in the microcirculation of the liver. The increased collagen deposition and fibrosis resulting from chronic hepatic inflammation increases hepatic stiffness and distorts the sinusoidal anatomy and vasculature [41].

There are also dynamic changes which occur in cirrhosis, leading to an increase in intrahepatic vascular tone. This is thought to contribute approximately 25% of the increase in vascular resistance [42]. Endothelial cells become dysfunctional, with reduced ability to produce nitric oxide in sufficient quantities to counteract the vasoconstrictive mediators released due to hepatic injury [41,43,44].

These changes lead to both fixed compression and narrowing of the lumen, as well as dynamic vasoconstriction with a resultant increase in vascular resistance [45,46,47]. Further compounding this problem is the potential to develop microthrombi from the reduction in the sinusoidal vascular calibre in combination with the inherent procoagulant state of cirrhosis, which can thereby further increase vascular resistance [45].

### 5.2. Causes of Increased Portal Blood Flow

Nitric oxide and other vasodilators produced by the intestinal microcirculation and in the splanchnic arterioles as a result of rising portal pressure also act on nearby blood vessels in the splanchnic circulation, causing splanchnic arterial vasodilation and increased inflow of blood into the portal system [45].

The pooling of blood in the splanchnic circulation reduces the effective circulating blood volume in the systemic circulation, activating neurohumoral systems including the renin–angiotensin—aldosterone system and antidiuretic hormones. This leads to sodium and water retention, expansion of the plasma volume and increased cardiac output, which further increases blood flow and perpetuates the higher portal pressures [44].

The pathophysiology of portal hypertension is shown in Figure 1.

## 6. Complications of Portal Hypertension and Their Role in Malnutrition and Sarcopenia (see Figure 2)

### 6.1. Portosystemic Collaterals

The portal vein is mainly supplied by the mesenteric and splenic veins, as well as other smaller tributary vessels, and terminates in the hepatic sinusoids. It effectively acts as a venous outflow system for the gut and carries blood and nutrients from the splanchnic vasculature to the liver for metabolism, storage and detoxification [48].

**Figure 2 nutrients-16-00035-f002:**
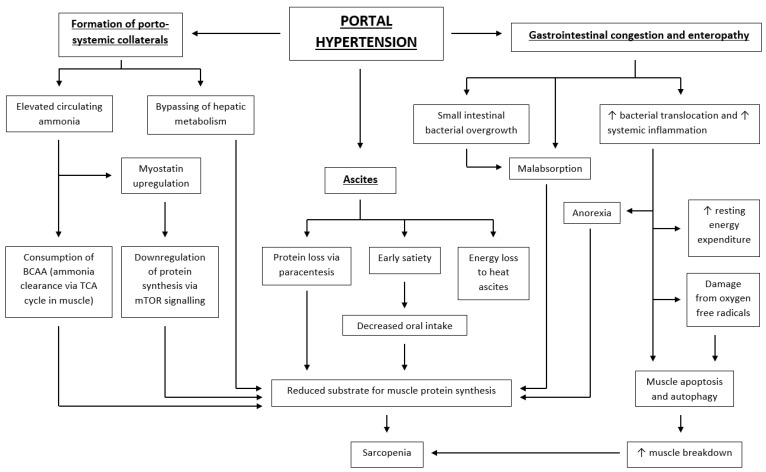
Complications of portal hypertension and their role in sarcopenia. Flowchart outlining how the complications of portal hypertension contribute to the development of sarcopenia. ↑ = increased; ↓ = decreased; BCAA = branched chain amino acid; TCA = tricarboxylic acid; mTOR = mammalian target of rapamycin.

In a patient with portal hypertension, elevations in vascular endothelial growth factor (VEGF), which is released by dysfunctional hepatic sinusoidal endothelial cells, HSCs and hepatocytes in response to inflammation [43], lead to the formation of collateral blood vessels that bypass the portal vein and decompress the portal system. These are recruited by recanalisation and dilatation of existing vessels to shunt blood from the portal system to the systemic circulation [41,49]. As portal hypertension progresses, increasing amounts of blood are shunted directly into the systemic circulation, bypassing the liver and the normal metabolism which occurs there.

Portosystemic shunting removes one of the major pathways by which ammonia is degraded and cleared by the body. Ammonia, a nitrogenous waste product, is normally converted via five enzymatic reactions in the mitochondria and cytosol of hepatocytes via the urea cycle (see Figure 3a) into the non-toxic waste product urea, which is then excreted by the kidneys. Increased shunting results in higher concentrations of ammonia reaching the systemic circulation, where it can cross the blood–brain barrier and cause neurotoxicity [50,51] and hepatic encephalopathy.

### 6.2. Muscle Impacts of Shunting of Ammonia-Rich Blood to the Systemic Circulation via Collaterals

The second pathway to clear ammonia is via the tricarboxylic acid (TCA) cycle in skeletal muscle [52]. The TCA cycle is the most important pathway for generating ATP for energy production for the body, via oxidation of acetylcoenzyme A (CoA) derived from carbohydrates, fatty acids and proteins (see Figure 3b). Alpha-ketoglutarate, a key intermediate of the TCA cycle, can detoxify ammonia by removing two molecules to form glutamine and glutamate [53]. This process depletes α-ketoglutarate, resulting in reduced ATP synthesis via the TCA cycle and subsequent mitochondrial dysfunction [54]. Branched chain amino acids (BCAAs) replenish α-ketoglutarate, but this leads to a reduction in circulating BCAAs that thereby deprives muscle cells of their preferred energy substrate [55]. Detoxification of ammonia in skeletal muscle also results in the production of reactive oxygen species, which can damage myocytes and include muscle autophagy [53].

Ammonia also acts to directly upregulate myostatin expression in muscle, which acts as a negative regulator of muscle cell growth and proliferation through the inhibition of intracellular signalling via the mammalian target of the rapamycin (mTOR) pathway [52,56]. This results in reduced skeletal muscle protein synthesis and further contributes to the impairment in ammonia clearance, due to reduced TCA cycle activity related to the reduction in muscle mass, and therefore a self-perpetuating cycle of muscle depletion.

In addition to the direct impacts of ammonia on muscle itself, the associated hepatic encephalopathy can impact nutrition by altering behaviours, including reducing oral intake and compliance with specific dietary advice. Groeneweg et al. also found that cirrhotic patients with subacute hepatic encephalopathy had significantly worse performance in ambulation and mobility, as well as eating, when compared to cirrhotic patients without encephalopathy [57]. This reduction in physical activity related to encephalopathy may further contribute to muscle wasting.

### 6.3. Impact of Systemic Inflammation on Muscle and Nutritional Parameters

Cirrhosis is a pro-inflammatory state, and portal hypertension is one of the major drivers of this systemic inflammation.

Firstly, portal hypertension has a direct impact on the integrity of the intestinal mucosal barrier and its function, causing vascular congestion and oedema of the gut wall, widening of the intercellular spaces and disruption of the tight junctions [58,59]. These structural changes increase intestinal permeability and allow for the translocation of bacteria and their products such as lipopolysaccharide, increasing the risk of infections such as spontaneous bacterial peritonitis and sepsis. Shunting of blood away from the liver via portosystemic collaterals further reduces the clearance of such pathogens by the liver.

In keeping with this, markers of bacterial translocation have been shown to be elevated in cirrhotic patients, with one study finding elevated lipopolysaccharide binding protein levels in 42% of cirrhotic patients with ascites [60]. Another study by Assimakopoulos et al. [61] found that plasma endotoxin levels were higher in patients with cirrhosis when compared to healthy controls, and that patients with decompensated cirrhosis had higher levels than those with compensated cirrhosis.

Bacterial translocation, even if transient, increases stimulation of the immune system via the intestinal lymphoid tissues, increasing activation of inflammatory cells such as monocytes, dendritic cells and lymphocytes [62]. The resultant release of pro-inflammatory cytokines promotes further upregulation of the immune system and maintains an inflammatory environment.

Multiple studies have analysed changes in systemic inflammatory markers in subjects with varying severities of liver disease [63]. In a study of 77 patients, including 14 healthy controls, 14 patients with compensated cirrhosis and 49 patients with decompensated cirrhosis, Dirchwolf et al. found that compared to the healthy controls, the patients with cirrhosis had higher levels of multiple inflammatory cytokines including IL-6, IL-7, IL-8, IL-12 and TNF-α, with IL-6 and IL-8 levels correlating with MELD score [64]. Giron-Gonzalez et al. also found higher serum TNF-α and IL-6 levels in 72 cirrhotic patients compared to 25 healthy controls, with the cytokine levels also being significantly higher in patients with Child–Pugh C cirrhosis compared to those with Child–Pugh A [65]. Salguero et al. also found that patients with Child–Pugh B HCV-related cirrhosis had higher levels of IL-6 and IL-10 compared to patients with Child–Pugh A cirrhosis [66].

The inflammatory cytokines and increased circulating NO also damage the gut wall and disrupt the tight junctions, further increasing intestinal permeability [67,68] and perpetuating the cycle of bacterial translocation and inflammation.

The deleterious impact of inflammation on muscle is due in part to the increased energy expenditure and metabolic demands associated with the inflammatory state. Activation of the immune system consumes a large amount of energy, up to 25% of the basal metabolic rate [69]. Once glycogen stores have been depleted, energy is supplied by fat breakdown as well as protein breakdown to supply amino acids as substrates for gluconeogenesis, with muscle being a preferential source [70]. In a study involving 25 patients with cirrhosis and 25 healthy controls, Glass et al. found that resting energy expenditure as measured by indirect calorimetry was significantly higher in the cirrhotic patients compared to healthy controls (66.7 ± 17.8 vs. 47.7 ± 7.9 kcal/cm^2^) [71].

Many inflammatory cytokines such as IFN, IL-1, IL-6 and TNF-α, also have an anorexigenic effect [72], leading to a reduction in oral intake. TNF-α induces and enhances expression of the leptin receptor, and IL-6 also promotes leptin secretion, resulting in decreased appetite. Increased leptin levels have also been associated with increased energy expenditure by increasing sympathetic nervous activity and activation of brown adipose tissue thermogenesis [73].

Pro-inflammatory cytokines also have a direct negative effect on muscle (see Figure 4). Cytokines including TNF-α, IL-1 and IL-6 promote muscle wasting via NF-KB [74], which upregulates myofibrillar proteolysis [75] and also induces muscle cell apoptosis [76]. Muscle protein synthesis is also affected [75], as IL-6 can induce insulin resistance, suppressing mTOR activity and reducing muscle synthesis [77]. These pro-inflammatory cytokines have also been shown to be elevated in the elderly—known as ‘inflammaging’—and have similarly been linked to sarcopenia [78,79] and frailty [80,81]. The impact of inflammation on sarcopenia is therefore four-fold, resulting in increased resting energy expenditure, reduced caloric intake, increased muscle apoptosis and downregulation of muscle protein synthesis.

### 6.4. Impact of Portal Hypertensive Enteropathy

The vascular congestion and oedema of the gastrointestinal tract due to portal hypertensive changes affects the stomach, small bowel and large bowel—called portal hypertensive gastropathy, enteropathy and colopathy, respectively. The reported prevalence of these portal hypertensive changes visible at endoscopy is variable but may be as high as 80% in patients with cirrhosis [82,83,84,85], with portal hypertensive gastropathy being the most frequently documented change.

Histological changes found in these conditions include congested and dilated capillaries and venules, and the thickening of the vessel walls, as well as mucosal oedema [86,87]. Fibromuscular proliferation in the lamina propria, thickening of the muscularis mucosae and a decreased villous/crypt ratio have also been described [86]; however, the significance of these changes when compared to patients without portal hypertension have not been replicated in all studies [88], and to date, there are no consensus diagnostic histologic criteria.

The changes in the mucosal microcirculation and the associated dysregulation can lead to mucosal hypoxia, resulting in the release and production of inflammatory cytokines and damaging oxygen free radicals which can further alter the integrity of the epithelium [89,90]. The damaged mucosa also has an impaired ability to heal itself [91] and is more susceptible to bleeding. Chronic occult bleeding from portal hypertensive gastropathy and enteropathy can result in iron deficiency.

Disruption to the mucosa and villi can lead to impaired absorption of both macro and micronutrients. Common micronutrient deficiencies in portal hypertension include vitamin A, vitamin D, magnesium, thiamine, vitamin B12, folic acid, zinc and selenium [11,20,92,93].

The clinical significance of protein-losing enteropathy (PLE) in malnutrition in cirrhosis and portal hypertension is not fully clear. While in theory, enteropathy can contribute to a loss of proteins and nutrients, the evidence in cirrhotic patients is conflicting. Case reports of PLE have been described as early as 1969 [94], where six patients with cirrhosis and portal hypertension were found to have elevated levels of enteric protein excretion, of which the two patients with the highest levels of excretion were also found to have intestinal lymphangiectasia. In a study including eleven patients with cirrhosis and portal hypertension, nine were found to have features of exudative enteropathy; however, protein loss did not correlate with the degree of portal hypertension [95]. Contrary to these findings, Georgopoulos et al. [96] did not find evidence of protein-losing enteropathy in sixteen patients with alcohol-related cirrhosis and portal hypertension.

Portal hypertension can also increase pressure within the intestinal lymphatic system, leading to dilatation, also known as lymphangiectasia. Disruption of the lymphatic system can lead to fat malabsorption and deficiency of the fat-absorbable vitamins A, D, E and K, which depend on the lymphatic capillaries for absorption and transport [97]. If the lymphangiectasia becomes severe enough, it can also lead to rupture, with leakage of the lymphatic fluid—which is rich in protein and lipoproteins—into the intestinal lumen, worsening losses of these nutrients; however, to date, this has only been described in case reports [98,99]. Further research is required to elucidate if this mechanism is currently under-recognised as a clinically significant contributor to malnutrition in cirrhosis and portal hypertension.

### 6.5. Disrupted Gastrointestinal Motility

Dysmotility has been commonly described in patients with cirrhosis. The exact underlying mechanism is not fully understood; however, autonomic dysfunction, which is most frequently manifested in the cardiovascular system, is thought to play a role. Possible contributors to autonomic dysfunction include vitamin E deficiency, elevated angiotensin-II and enhanced endothelial nitric oxide, resulting in defective vascular responsiveness to noradrenaline [100].

All parts of the gastrointestinal tract can be affected. Numerous studies have reported delayed gastric emptying in patients with cirrhosis and portal hypertension compared to healthy controls [101,102,103]. Intestinal motility can be affected, with altered timing, frequency and amplitude of contractions described [104,105,106], with cirrhotic patients with portal hypertension being more frequently affected than those without portal hypertension [105]. Numerous studies have looked at intestinal transit times in patients with cirrhosis and portal hypertension, with most finding prolonged transit times [107,108,109]; however, several studies have found accelerated transit times through the proximal small intestine [110] or colon [111]. Accelerated intestinal transit can increase nutrient malabsorption [112].

Decreased gastric motility can result in early satiety, abdominal pain and decreased oral intake. Delayed intestinal transit can also lead to small intestinal bacterial overgrowth (SIBO) [113,114]. SIBO can worsen bacterial translocation [109] and also contribute to malabsorption by causing damage to the intestinal membrane either by direct adherence and disruption of the membrane or by the production of enterotoxins. Bacteria can also cause intraluminal degradation of carbohydrates and proteins, impairing absorption via the usual processes and resulting in symptoms such as bloating from excess methane and gas production, abdominal pain and diarrhoea [115]. Increased protein degradation can also lead to increased ammonia production by the bacteria, increasing the risk of hepatic encephalopathy. SIBO can also lead to malabsorption through the bacterial deconjugation of bile acids, depleting the bile acid pool and leading to fat malabsorption [116].

### 6.6. Ascites

Splanchnic dilatation and pooling of blood in the portal system results in a relative hypovolaemia of the systemic circulation and hypoperfusion of the kidneys, activating the renin–angiotensin–aldosterone system and causing renal vasoconstriction and fluid retention. The retained fluid increases blood volume and hydrostatic pressure in the blood vessels. This, in combination with the increased vascular wall permeability and decreased oncotic pressure as a result of hypoalbuminemia, can lead to fluid leaking out into the interstitial space and peritoneal cavity, forming ascites [117].

Large volume ascites can compress and physically limit the capacity of the stomach, resulting in early satiety and decreased oral intake. Patients are often able to consume larger meals after large volume paracentesis, as reported by Aqel et al., who found that patients had a higher fasting gastric volume (241 mL to 312 mL), tolerated larger volume meals (738 mL to 964 mL) and increased their caloric intake by 11.6% (3110 kcal to 3470 kcal) post-paracentesis [118]. However, in patients with refractory ascites, gastric reserve may again decrease as the ascites reaccumulates.

Ascitic fluid also contains nutrients such as fats, carbohydrates and proteins [119], which are removed during large volume paracentesis. Administration of albumin replaces the fluid loss by promoting plasma volume re-expansion but does not replace the energy and nutrient losses, resulting in a net energy loss with each paracentesis.

Ascites can also theoretically increase resting energy expenditure due to the energy required to heat and maintain the fluid at body temperature; however, there is limited evidence to support this. In a small study involving ten patients with cirrhosis and ascites, Dolz et al. found that resting energy expenditure decreased significantly from 1682 ± 291 kcal/day to 1523 ± 240 kcal/day following paracentesis [120]; however, these findings were not replicated by Knudsen et al., who found no difference after paracentesis in 19 cirrhotic patients [121].

## 7. Treatment of Portal Hypertension

Many of the treatment options for patients with cirrhosis and portal hypertension are currently targeted at managing the specific clinical complications. For example, diuretics are administered for ascites and lactulose or rifaximin for hepatic encephalopathy. However, there are now several specific therapies that target portal hypertension itself.

Several treatment strategies have been shown to ameliorate sarcopenia in patients with cirrhosis such as testosterone therapy [122,123] and nutritional interventions such as BCAA supplementation [124,125]; however, these do not directly pertain to portal hypertension and are outside the scope of this paper.

### 7.1. Treatment of Underlying Aetiology of Liver Disease

Treatment of the underlying aetiology of liver disease can improve portal hypertension, although there is no study that directly links reduced portal pressure to improved nutrition. Successful eradication of chronic hepatitis C infection can reduce HVPG by up to 30% [126,127], and viral suppression of chronic hepatitis B infection can reduce HVPG by up to 20% [128]. Similarly, sustained abstinence of alcohol can significantly reduce HVPG by up to 46% [129,130]. Anticoagulation for cirrhotic patients with portal vein thrombosis (PVT) decreases the progression of thrombosis and increases recanalisation, theoretically reducing HVPG. One meta-analysis of 353 patients in eight studies reported that 71% of patients with PVT treated with anticoagulation achieved recanalisation compared to only 42% of patients who were not treated [131].

The role of anticoagulation in patients without portal vein thrombosis is more controversial due to the concerns of bleeding in these patients. Villa et al. found that a 12-month course of 4000 units/day of enoxaparin was safe and effective in preventing portal vein thrombosis (PVT) in a cohort of 70 patients with Child–Pugh B and C cirrhosis, with no patients developing PVT compared to 16.6% of control patients during the treatment period [132], although end-of-study HVPG was not measured. Decompensation of liver disease was also lower in the enoxaparin-treated group (38.2% vs. 83%, *p* < 0.001). A possible explanation for this is that enoxaparin treatment prevents micro- and macrothrombi in the portal system, thereby decreasing intrahepatic resistance and reducing portal pressure; however, further studies are warranted to quantify the effect of anticoagulation on portal pressures and to assess its possible impact on nutrition and sarcopenia.

### 7.2. Beta-Blockers

Blockage of β1 receptors (located in cardiac muscle) and β2 receptors (located in splanchnic vessels) results in decreased cardiac output and splanchnic vasoconstriction and decreased splanchnic blood flow, which helps lower portal pressures. The aim of non-selective beta-blocker (NSBB) use is to reduce the hepatic venous pressure gradient by >20% or to <12 mmHg. The traditionally used NSBB in cirrhotic patients was propranolol, which acts on both β1 and β2 receptors. The preferred agent has become carvedilol in recent years, which also blocks α1-adrenergic receptors, decreasing both portocollateral and intrahepatic resistance, and creating a more marked decrease in portal pressure than propranolol [2,133,134].

NSBBs are commonly employed to reduce the risk of variceal bleeding in cirrhotic patients, but previous studies have also demonstrated reduced intestinal permeability and bacterial translocation with propranolol in patients with cirrhosis [135,136]. Several studies have shown attenuation of muscle mass loss with beta-blockers in patients with cancer-related cachexia [137] and in patients with heart failure [138]; however, there are no dedicated prospective studies assessing the impact of NSBBs on nutritional status and sarcopenia in patients with cirrhosis and portal hypertension. A retrospective study by Li et al. compared 20 cirrhotic patients treated with NSBBs with 20 propensity-score matched cirrhotic patients not on NSBBs who had abdominal CTs after 4 years of follow up. The cirrhotic patients who did not receive NSBBs had a significantly reduced skeletal muscle index at 4 years (mean SMI alteration −1.782 ± 4.624 cm^2^/m^2^) compared with patients who did receive NSBBs (mean alteration 1.195 ± 5.633 cm^2^/m^2^) [139]. Further, ideally, prospective studies are needed to clarify the effect of NSBBs on malnutrition and sarcopenia in patients with cirrhosis and portal hypertension.

### 7.3. TIPS

The transjugular intrahepatic portosystemic shunt (TIPS) is a radiologically inserted stent which creates a channel between the portal vein (inflow) and the hepatic vein (outflow), allowing blood to bypass the liver and be shunted directly into the systemic circulation, decompressing the portal system. The most common indications for TIPS currently are the management of refractory ascites and hepatic hydrothorax, Budd–Chiari syndrome, and for variceal haemorrhage [2,140,141].

The resolution of ascites after TIPS insertion has been shown to result in improvements in nutritional parameters such as dry weight, total body fat and BMI in patients with cirrhosis in some studies [142,143]. TIPS also improves nutritional status in patients with non-cirrhotic portal hypertension [144]. There have also been case reports of protein-losing enteropathy resolving after the placement of TIPS [145].

TIPS has also been shown to increase muscle mass, as quantified by CT scan. In several retrospective studies involving between 57 and 224 cirrhotic patients undergoing TIPS, muscle mass was shown to increase after as little as three months [146], with the biggest changes seen after longer follow up [147,148]. Importantly, increased muscle mass post-TIPS stent has been shown to correlate with improved survival, reiterating the importance of treating sarcopenia in patients with portal hypertension [148]. In a prospective cohort study involving 12 patients, Hey et al. found that muscle mass significantly improved at six months post-TIPS insertion, with the mean skeletal muscle area increasing from 139.32 ± 22.72 cm^2^ to 154.64 ± 27.43 cm^2^ [149].

Muscle function, however, did not improve in the study by Hey et al., which may reflect the dual action of the TIPS stent, in that it does not only decompress portal pressures but also increases shunting of ammonia-rich portal blood directly into the systemic circulation. This increased circulating ammonia can also lead to hepatic encephalopathy, one of the main adverse effects of TIPS, which as mentioned above, may worsen nutrition due to decreased oral intake from confusion and altered behaviour, as well as having a deleterious effect on muscle.

### 7.4. Terlipressin

Terlipressin is a vasopressin analogue derived from the hormone lysine-vasopressin. It acts predominantly on the V1a receptors located on vascular smooth muscle in the splanchnic circulation and mediates potent vasoconstriction in the splanchnic and systemic arteriolar circulation [150]. This results in reduced portal venous inflow and reduced portal pressures [151].

In a cohort of 16 patients with cirrhosis, 11 of whom had a baseline HVPG of >12 mmHg, Møller et al. found that the median HVPG fell significantly from 16.8 to 12 mmHg after administration of 2 mg terlipressin [150]. Another study by Escorsell et al. also found a significant reduction of 20.7 ± 11.5% in HVPG after 2 mg terlipressin and a reduction of 15.5 ± 8.8% after 1 mg terlipressin, compared to no significant changes after placebo in a cohort of 23 patients [152].

International guidelines currently only recommend the use of terlipressin for the management of acute variceal haemorrhage and hepatorenal syndrome [2]. More recently, the effect of terlipressin on other complications of portal hypertension has been explored in research studies including its impact on sarcopenia.

Several studies have looked at the effect of terlipressin on ascites and hepatic hydrothorax. In a small study involving five patients with cirrhosis and diuretic-refractory ascites from our institution, 4 weeks of terlipressin treatment significantly reduced paracentesis volume from a median of 22.9 L to 11.9 L [153]. A later study demonstrated that the frequency of large volume paracentesis significantly decreased by 62% in 23 patients who were treated with terlipressin for a median duration of 51 days [154]. A study by Bajaj et al. involving six patients found that terlipressin treatment resulted in a ≥30% reduction in ascites volume drained during paracentesis in all patients, and a ≥50% increase in the interval between large volume paracentesis was reported in four of the six patients [155].

There is evidence for the use of continuous terlipressin infusions in the outpatient setting, with our institution reporting our experience of 23 patients receiving ambulatory terlipressin, while on the waitlist for a liver transplant, for the management of hepatorenal syndrome, refractory ascites or hepatic hydrothorax for a combined total of 2844 days. These data showed not only improvements in renal function and ascites but importantly, no major drug-related adverse events [154]. A larger patient experience of >100 patients and 12,000 patient days confirming these safety data has been reported in abstract form [156] and represents the potential for this therapy to be continued in the long term.

Data from our centre describing the impact of home terlipressin on nutritional and muscle parameters demonstrate its potential dramatic impact on a sequela of cirrhosis, for which an inexorable decline is usually observed. While in observation cohorts, handgrip strength usually declines by 0.13–0.23 kg per month [157,158], our data suggest a powerful treatment effect of terlipressin, with a 3 kg improvement in handgrip strength, from 25.36 ± 8.13 kg to 28.49 ± 7.63 kg, seen in a study of 19 patients with cirrhosis and sarcopenia treated with terlipressin for a median of 51 days [159].

There is also emerging evidence that prolonged use of terlipressin can lead to improved nutritional parameters. In the same study which found improved handgrip strength, Chapman et al., also found that treatment with terlipressin led to a significant improvement in energy (17.94 ± 5.42 vs. 27.70 ± 7.48 kcal/g) and protein (0.74 ± 0.28 vs. 1.16 ± 0.31 g/kg) intake, which may relate to improvements in ascites volume [159].

Currently, most data regarding long-term terlipressin come from our own institution, with relatively small patient numbers, and further multicentre studies are required to confirm these findings before they can be routinely recommended for these indications in clinical practice guidelines.

### 7.5. Statins

Statins, which are HMG-CoA reductase inhibitors, have been found to upregulate nitric oxide and downregulate hepatic stellate cell activation, decreasing hepatic fibrogenesis and improving endothelial function and intrahepatic vascular resistance [160,161]. Statins have also been shown to have anti-inflammatory properties by inhibiting the binding of ICAM-1, resulting in decreased leucocyte adhesion to the endothelium and decreasing the production of NF-kB and the release of pro-inflammatory cytokines such as IL-6 and TNF-α [162,163].

There are currently only a limited number of studies evaluating the effect of statins in patients with cirrhosis and portal hypertension. In three randomised control studies involving a total of 102 cirrhotic patients, statins were found to reduce portal pressure by at least 20% or to <12 mmHg in 30–90% of patients [164,165,166]. In two of these trials, the statin was coadministered with NSBBs, and the combination of statin and NSBBs resulted in a higher reduction in HVPG than in patients who received NSBBs alone [164,166]. In several other retrospective observational studies, statins were noted to reduce the risk of decompensation and improve mortality in cirrhotic patients [167,168,169]; however, most of the patients included in the trials were patients with compensated Child–Pugh A cirrhosis. Unfortunately, none of these studies measured nutritional or muscle outcomes, and further studies are needed to evaluate the potential impact of statin therapy on sarcopenia in this cohort.

### 7.6. Exercise

Physical exercise has been shown to improve muscle strength and function in the general population, including in frail elderly patients [170]; however, data in patients with cirrhosis are more limited due to safety concerns, and few studies have looked at the effect of exercise interventions on portal pressure in these patients.

Macias-Rodriguez et al. [171] conducted a randomised control trial comparing 14 weeks of moderate intensity exercise performed via three supervised exercise sessions per week, involving 40 min of aerobic exercise via cycle ergometry and 30 min of kinesiotherapy (*n* = 13), to a control arm receiving nutritional therapy only (*n* = 12). After 14 weeks, the HVPG decreased by 2.5 mmHg from baseline in the exercise therapy group and increased by 4 mmHg in the control group (*p* = 0.009).

Berzigotti et al. [172] enrolled 60 patients in a prospective uncontrolled study to receive an intensive 16-week lifestyle intervention program with a personalised hypocaloric normoproteic diet and a 60 min per week session of supervised gym-based aerobic and resistance exercise. In this cohort, the HVPG decreased by 1.6 mmHg from 13.9 ± 5.6 to 12.3 ± 5.2 mm Hg (*p* < 0.0001), with 42% of patients experiencing a reduction in HVPG of ≥10%. Body composition also changed, with a 5 kg decrease in body weight and significantly decreased fat mass and waist circumference observed; however, there was no change in lean mass.

Due to the limited studies published in the literature which have a high degree of heterogeneity in intervention types and outcome measures, there is currently no standardised recommendation regarding the type, intensity and duration of exercise in patients with cirrhosis.

## 8. Conclusions

Malnutrition and sarcopenia are common in patients with cirrhosis, increase in prevalence with increasing severity of liver disease and have a significant impact on morbidity and mortality. This review summarises the pivotal role played by portal hypertension in the development of malnutrition in cirrhosis. Limited data suggest that active therapies to specifically combat portal hypertension may be able to reverse the sarcopenia and nutritional compromises related to cirrhosis, which is particularly pertinent to those with decompensated cirrhosis undergoing consideration for liver transplant. Early data on terlipressin therapy are particularly promising, with some conflicting results reported following TIPS insertion. Further large-scale studies specifically designed with nutritional and muscle outcomes in mind are required to examine interventions that combat portal hypertension in this population.

## Figures and Tables

**Figure 1 nutrients-16-00035-f001:**
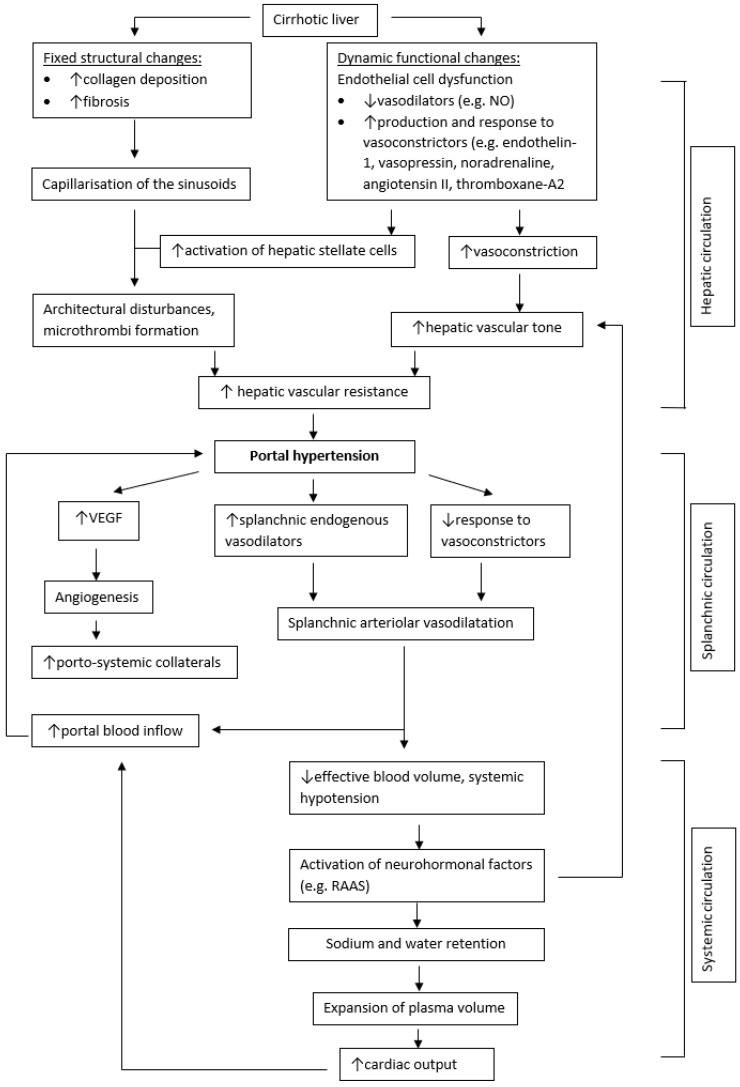
Pathophysiology of portal hypertension. Flowchart describing the development of portal hypertension in cirrhosis. ↑ = increased; ↓ = decreased; NO = nitric oxide; VEGF = vascular endothelial growth factor; RAAS = renin–aldosterone–angiotensin system.

**Figure 3 nutrients-16-00035-f003:**
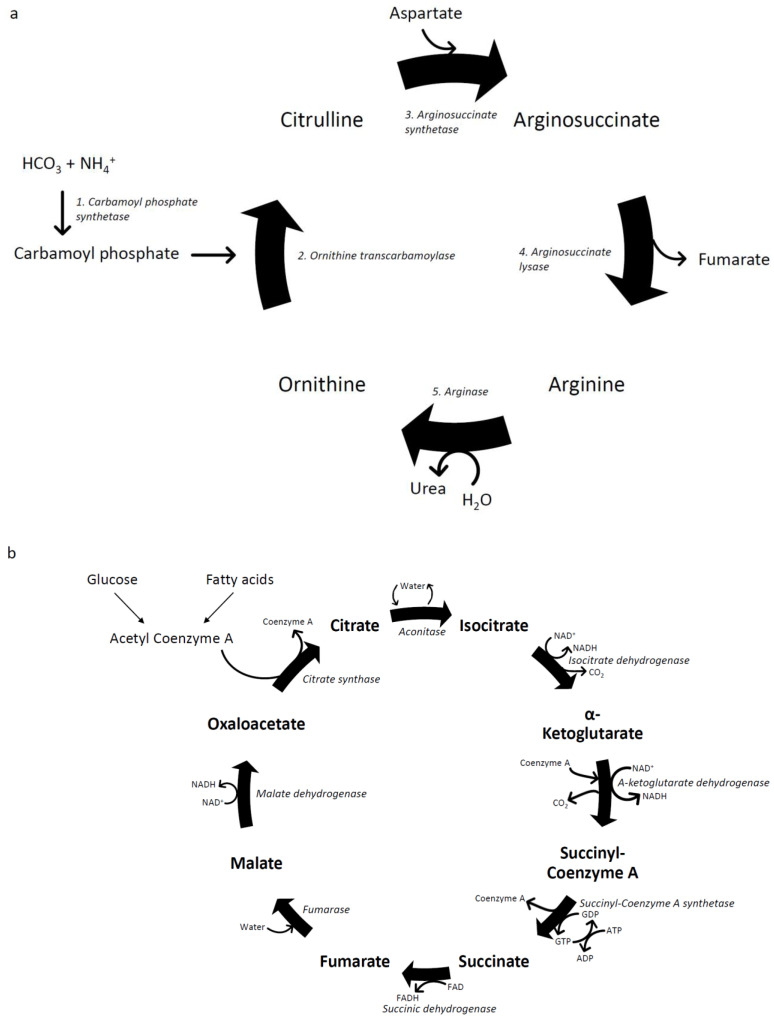
The urea cycle and the tricarboxylic acid cycle. (**a**) Flowchart describing the steps of the urea cycle; (**b**) flowchart describing the steps of the tricarboxylic acid cycle.

**Figure 4 nutrients-16-00035-f004:**
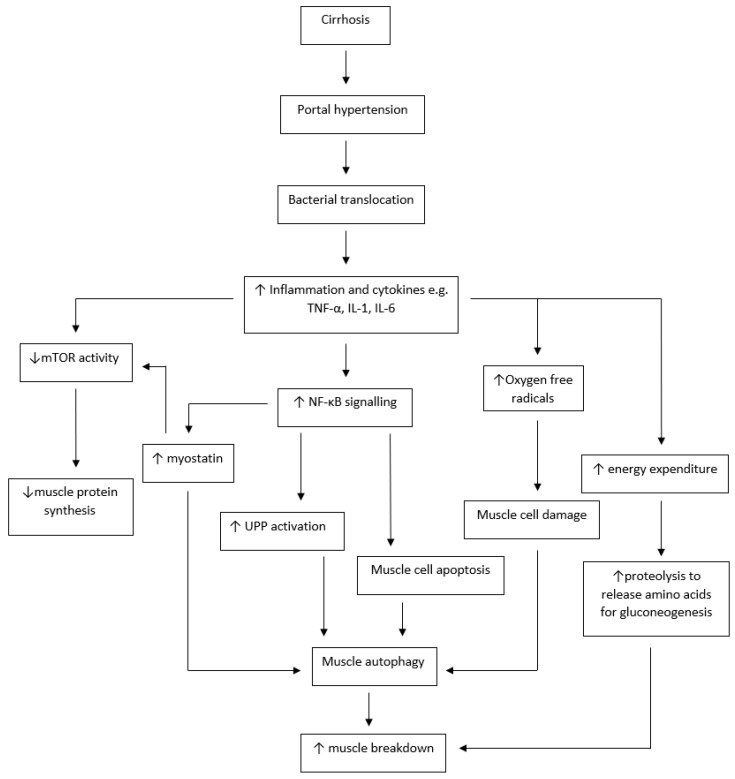
The effect of inflammation on muscle. Flowchart outlining the effect of inflammation on muscle. ↑ = increased; ↓ = decreased; mTOR = mammalian target of rapamycin; UPP = ubiquitin proteasome pathway.

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
