# Peer review of "Portal Hypertension in Malnutrition and Sarcopenia in Decompensated Cirrhosis—Pathogenesis, Implications and Therapeutic Opportunities"

_nutrients, 2023, doi:10.3390/nu16010035_

Round 1

Reviewer 1 Report

Comments and Suggestions for Authors

The authors do a commendable job of providing an insightful review of how portal hypertension contributes to the development of sarcopenia in cirrhosis. They do a laudable job in reviewing the current treatment strategies to tackle portal hypertension and sarcopenia in cirrhosis.

A few minor specific comments

Figure 1: Cirrhosis at the top seems to be hanging without any connections.

Figure 2: The figure is so busy and so hard to understand. It lacks a pivot from which to follow. Creating a logical flow will be very helpful.

Line 205: Is it ATP or APT?

Line 272: Typically, once glycogen stores are depleted, the next energy supply would be from fat breakdown before protein breakdown. What is the reason for skipping fat breakdown?

Line 394: It seems like something is missing. “...following paracentesis from???”

Line 412: It seems the acronym PVT is first utilized here. I suppose its full form should be provided here.

Author Response

We thank the reviewer for their thoughts and feedback on our paper. Please find below our response to the specific comments and questions raised by the reviewer:

A few minor specific comments 

Figure 1: Cirrhosis at the top seems to be hanging without any connections. 

The figure has been amended to add arrows connecting cirrhosis to the fixed and dynamic changes.

Figure 2: The figure is so busy and so hard to understand. It lacks a pivot from which to follow. Creating a logical flow will be very helpful. 

The figure has been amended to create a more logical flow that is easier to follow.

Line 205: Is it ATP or APT? 

Thank-you for picking up this spelling error. This has been corrected to ATP.

Line 272: Typically, once glycogen stores are depleted, the next energy supply would be from fat breakdown before protein breakdown. What is the reason for skipping fat breakdown? 

Thank-you for raising this vaild point. We have amended the manuscript to add that once glycogen stores have been depleted, energy is supplied from both lipolysis and proteolysis.

Line 394: It seems like something is missing. “...following paracentesis from???” 

Thank-you for picking up this grammatical error. The extra “from” has been deleted.

Line 412: It seems the acronym PVT is first utilized here. I suppose its full form should be provided here. 

Thank-you for picking up this oversight. Portal vein thrombosis is first used in full in line 411 (of the original manuscript). The acronym has now been added in parentheses after this first use. 

Reviewer 2 Report

Comments and Suggestions for Authors

It is an interesting study about the association between portal hypertension and sarcopenia in cirrhotic patients.  I have some minor comments to make 

1. In the section of prevalence of sarcopenia, please add data about the role, of ultrasound on the diagnosis of sarcopenia 

2 The section regarding the pathophysiology of portal hypertension contains details about the pathogenesis of portal hypertension that are useless and out of the scope of this review. The whole section must be removed  (lines 110-169)

3. Please give information about the role of cirrhotic cardiomyopathy on the pathogenesis of sarcopenia 

4. At the section of treatment, you describe how treatment of the underlying causative factor of liver disease may improve portal pressure.  Moreover you analyze the association between anticoagulants and improvement of  portal pressures . Nevertheless,  you do not give information whether any decline in portal pressure has been associated with improvement of sarcopenia 

5. Please clarify that terlipressin is not indication for treatment of portal pressure or sarcopenia. Make it more clear that terlipressin could not be recommended for treatment of sarcopenia 

6. Add a paragraph about the role of iv human albumin on treatment of sarcopenia 

7. Add a paragraph about the role of nutrition protocols and exercise on treatment of sarcopenia 

Comments on the Quality of English Language

There are some grammar and syntax errors

Author Response

We thank the reviewer for their thoughts and feedback on our paper. Please find below our response to the specific comments and questions raised by the reviewer:

1. In the section of prevalence of sarcopenia, please add data about the role, of ultrasound on the diagnosis of sarcopenia 

Ultrasound has been added as a diagnostic method, however as mentioned in the article, the strengths and weaknesses of each diagnostic method have been well described elsewhere and are therefore not the main aim of the article. We have therefore not included detailed information on its role in diagnosing sarcopenia.

2. The section regarding the pathophysiology of portal hypertension contains details about the pathogenesis of portal hypertension that are useless and out of the scope of this review. The whole section must be removed  (lines 110-169)

We have removed the section on normal liver haemodynamics. We have kept the section on increased intrahepatic resistance and increased portal flow as we believe they are relevant to the therapeutic options discussed later in the paper, however have made this section more succinct.

3. Please give information about the role of cirrhotic cardiomyopathy on the pathogenesis of sarcopenia 

The incidence of cirrhotic cardiomyopathy has been shown to be higher in patients with sarcopenia, however as yet there is no evidence to show that cirrhotic cardiomyopathy causes sarcopenia. As the purpose of this paper is to focus on the causal link between portal hypertension and sarcopenia, we feel that cirrhotic cardiomyopathy, which is a large topic in itself, is outside the scope of this paper, therefore have not included it.

4. At the section of treatment, you describe how treatment of the underlying causative factor of liver disease may improve portal pressure.  Moreover you analyze the association between anticoagulants and improvement of  portal pressures . Nevertheless,  you do not give information whether any decline in portal pressure has been associated with improvement of sarcopenia 

Few studies have examined the link between quantified portal pressures and sarcopenia, and there is no literature to our knowledge that directly links falling portal pressure to improving nutrition, but this would be an interesting subject of future research. We have added a line to state that there is no data that links falling portal pressure to improved nutrition.

5. Please clarify that terlipressin is not indication for treatment of portal pressure or sarcopenia. Make it more clear that terlipressin could not be recommended for treatment of sarcopenia 

We have clarified that current international guidelines currently only recommend the use of terlipressin for the management of acute variceal haemorrhage and hepatorenal syndrome.

We have already previously noted that the data for terlipressin in malnutrition and sarcopenia is from our own institution, and that further multi-centre studies are required before terlipressin can be recommended for these indications.

6. Add a paragraph about the role of iv human albumin on treatment of sarcopenia 

IV albumin does not appear to lead to a reduction in portal pressures1, and is also not established as a therapy for sarcopenia. We have therefore not included it as a treatment option in this review, as its multiple other benefits including anti-inflammatory effects are beyond the scope of this paper.

1. J. Fernández et al., "Effects of Albumin Treatment on Systemic and Portal Hemodynamics and Systemic Inflammation in Patients With Decompensated Cirrhosis," Gastroenterology, vol. 157, no. 1, pp. 149-162, 2019.

7. Add a paragraph about the role of nutrition protocols and exercise on treatment of sarcopenia 

Thank-you for this comment. We have added a paragraph discussing the studies relating exercise to a reduction in portal hypertension.

There is little evidence relating diet to portal pressures, however we acknowledge that nutritional protocols and testosterone therapy can ameliorate sarcopenia in these populations.  We have added a line on "other" interventions at the start of the treatment section, acknowledging some treatment strategies for sarcopenia in cirrhosis that do not directly pertain to portal hypertension.